# Multiomic analysis of S*chistosoma mansoni* reveals unique expression profiles in cercarial heads and tails

James R. Hagerty [1], Hyung Chul Kim[1] & Emmitt R. Jolly [1,2✉]

Schistosomes require both molluscan and mammalian hosts for development. The larval cercaria exits the snail host and swims to identify and invade the mammalian host. The cercaria has two macrostructures, the head and the tail. The head invades the host, where it matures into an adult worm. The tail is lost after host invasion. Translation in the cercaria differs in each macrostructure, with higher levels of translation in the cercarial tail and little to no translational activity in the cercarial head. We compared the transcriptome and proteome of the cercarial head and tail and observed stark differences between the two macrostructures. We identified unique and differentially expressed transcripts and proteins, including ribosomal components expressed in higher levels in tails than in heads, which may explain the differences in translation levels between heads and tails. We also characterized the weak correlation between transcription and translation in infectious cercarial heads and tails.

[1] Case Western Reserve University, Department of Biology, Cleveland, OH, USA. [2] Case Western Reserve University, Center for Global Health and Disease, Cleveland, OH, USA. ✉email: emmitt.jolly@case.edu

Schistosomes have a complex life cycle that must quickly adapt both physiologically and morphologically to distinct environments. The schistosome infectious larval stage, cercaria, exits the intermediate molluscan host in search of a definitive mammalian host. After identifying and penetrating the mammalian host, the cercaria loses its tail, transforms into a schistosomulum, and enters the host circulatory system. Over several weeks, the schistosomulum develops into an adult worm in the mesentery of the liver, pairs with a mate, and produces hundreds of eggs daily. The eggs are excreted out of the host and hatch into transient miracidia. The miracidia then invade a freshwater snail and develops into mother and daughter sporocysts. The sporocyst produces and releases the infectious cercariae, completing the life cycle. The cercaria is transiently free-living and represents the first interaction point in the parasite life cycle with the human host. This cercarial stage has been studied using several -omic approaches, including microarrays, RNA-seq, and proteomics; however, our understanding of this stage's transcriptional and translational control mechanisms is still limited[1–8]. Collectively, -omic approaches to date have consistently pointed to a substantial upregulation of genes related to glycolysis and metabolism[9], consistent with the functional activity of actively swimming cercaria in search of a mammalian host. Proteomic analysis of the whole cercariae and the cercarial secretions released from the cercarial head revealed an abundance of proteases, including elastase and other proteins that are likely involved in host invasion[4,9–11]. The cercarial tail, however, is primarily involved in motility. Given the abundance of proteases in the translation-limited cercarial head and the relatively translation-enhanced cercarial tail, the combinational analysis of the two structures potentially clouds our understanding of each macrostructure[12,13].

Previously, we proposed treating the free-swimming cercarial stage as two separate -omic entities due to the apparent differences in translational regulation and biological role of the cercarial macrostructures: head and tail[13]. The cercarial head had significantly lower global translation levels than the cercarial tail. Here, we add to these initial observations and have analyzed and compared the transcriptome and proteome of the cercarial head and tail. Analysis of the head and tail as separate structures has allowed us to elucidate potential mechanisms for regulating the observed translational differences and functional roles of heads and tails. We found that cercarial heads and tails: (1) store distinct populations of proteins and transcripts which correlate to their functional roles as macrostructures, (2) differentially regulate translation using ribosomal component composition, and (3) have a weak correlation between transcript and protein abundance. These findings demonstrate the necessity for treating cercariae as two distinctive macrostructures instead of single units for study.

## Results

### Cercarial heads and tails are transcriptomically and proteomically distinct.
The functional and developmental roles for cercarial heads and tails are distinct. The cercarial tail gives motility and assists in host penetration; however, it is discarded and does not progress through development in the definitive host. In contrast, the cercarial head develops into the adult worm and stores all the necessary transcripts and proteins necessary for initial entry and adaptation to the definitive host[14,15]. Since cercariae do not undergo transcription[14], and translation in cercarial heads and tails is differentially regulated[13], we explored the conservation of steady-state transcript populations between cercarial heads and tails.

We found a total of 12,533 transcripts in cercariae (Fig. 1a). Cercarial heads and tails share 5,312 transcripts. Among these shared genes are four of the most abundant transcripts in cercariae: *Calmodulin-4 Smp_032990*, *Calcium-binding protein Smp_033000*, and two uncharacterized genes *Smp_195070* and *Smp_318920* (Supplementary Data 1)[16,17]. Cercarial heads have 207 unique transcripts, and cercarial tails have a large store of 7,014 unique transcripts (Fig. 1a). DESeq2 utilizes a Log fold change (LFC) Shrinkage process that reduces the noise created by low abundance transcripts or transcripts only detected in one macrostructure allowing for effective relative comparison of low abundance transcripts[18].

We next investigated protein storage in cercarial heads and tails using label-free mass spectrometry. We identified 2401 proteins and were able to verify 684 of those against the proteins identified by Sotillo et al. (Supplementary Data 4)[19]. Cercarial heads and tails share 1,856 of the 2,401 identified proteins. Four hundred and fifty-six (456) proteins were only identified in ≥ 2 replicates of cercarial heads, and 89 were only identified in two or more replicates in cercarial tails. Within the population of proteins fully identified only in heads or tails, 136 unique head proteins were identified in ≥2 replicates in cercarial heads and were undetected in cercarial tails (Fig. 1b). We also identified 43 unique tail proteins that were identified in ≥2 replicates in cercarial tails and were undetected in cercarial heads (Fig. 1b). However, while tails have fewer unique proteins compared to heads, they have a higher number of overall transcripts. Unique proteins have been identified in one macrostructure but were not detected in the other macrostructure. The uniquely identified proteins were not analyzed for relative differential expression and are reported separately from our expression analysis and are removed during the filtering in the spectral matching process of label-free quantitation. These proteins are still of interest because they are only stored at detectable levels in a single macrostructure, either the head or the tail. Correspondingly, cercarial tails store a broader diversity of transcripts, while cercarial heads store a broader diversity of proteins.

We used differential expression analysis to look for variation in the transcriptomes and proteomes of heads and tails. In total, 4,511 transcripts were differentially regulated (log2FoldChange ≥ 2.0 and p-value ≤ 0.01) out of 12,533 transcripts identified in both heads and tails (Fig. 2a). Of the 4,511 differentially regulated transcripts, 164 were upregulated in cercarial heads, and 4,347 were upregulated in cercarial tails (Fig. 2a, Supplementary Data 1). To quantify the differential expression of proteins in cercarial heads and tails, we utilized label-free mass spectrometry, identifying a total of 463 differentially expressed proteins (Fig. 2b). The distribution of differential protein expression between cercarial heads and tails is relatively even. Cercarial heads have 255 upregulated proteins, and cercarial tails contain 208 upregulated proteins (Fig. 2b).

### Cercarial heads store ribosomal transcripts, while cercarial tails store ribosomal proteins.
Given the different functional roles of cercarial heads (host invasion), and cercarial tails (motility and metabolism), we predicted a varied representation in both unique (identified only in head or only in tail samples) and differentially expressed transcripts and proteins in the macrostructures. We performed a gene ontology (GO) enrichment analysis to identify enrichment and reduction of transcript and protein types in 3 categories: biological process (BP), molecular function (MF), and cellular compartment (CC). The enriched status is reported for gene ontology categories if the population of transcripts or proteins that match that category is higher than expected by chance. The reduced status for a GO category is reported if the number of transcripts or proteins that match that category is less than what is expected by chance. To account for false positives and false

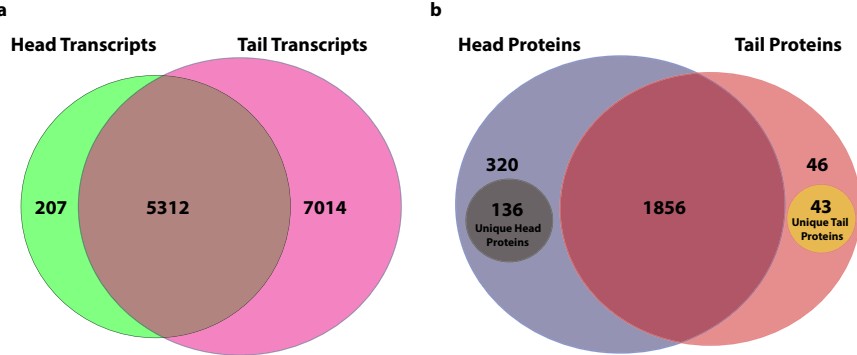

**Fig. 1 Total identified proteins and transcripts in cercarial heads and tails. a** Venn Diagram showing all transcripts identified above the ≥1.0 normalized count threshold in one or both macrostructures. **b** Venn Diagram showing proteomic identification of proteins with ≥2 unique peptides, identified in ≥2 replicates. **b** includes sub-populations of unique head and unique tail proteins identified in ≥2 replicates in one macrostructure and 0 replicates in the other macrostructure. One hundred and thirty-six (136) unique head proteins were identified only in cercarial heads, and 43 unique tail proteins were identified only in cercarial tails. All supporting data is contained in Supplementary Data 1. Replication $n = 3$ independent biological replicates.

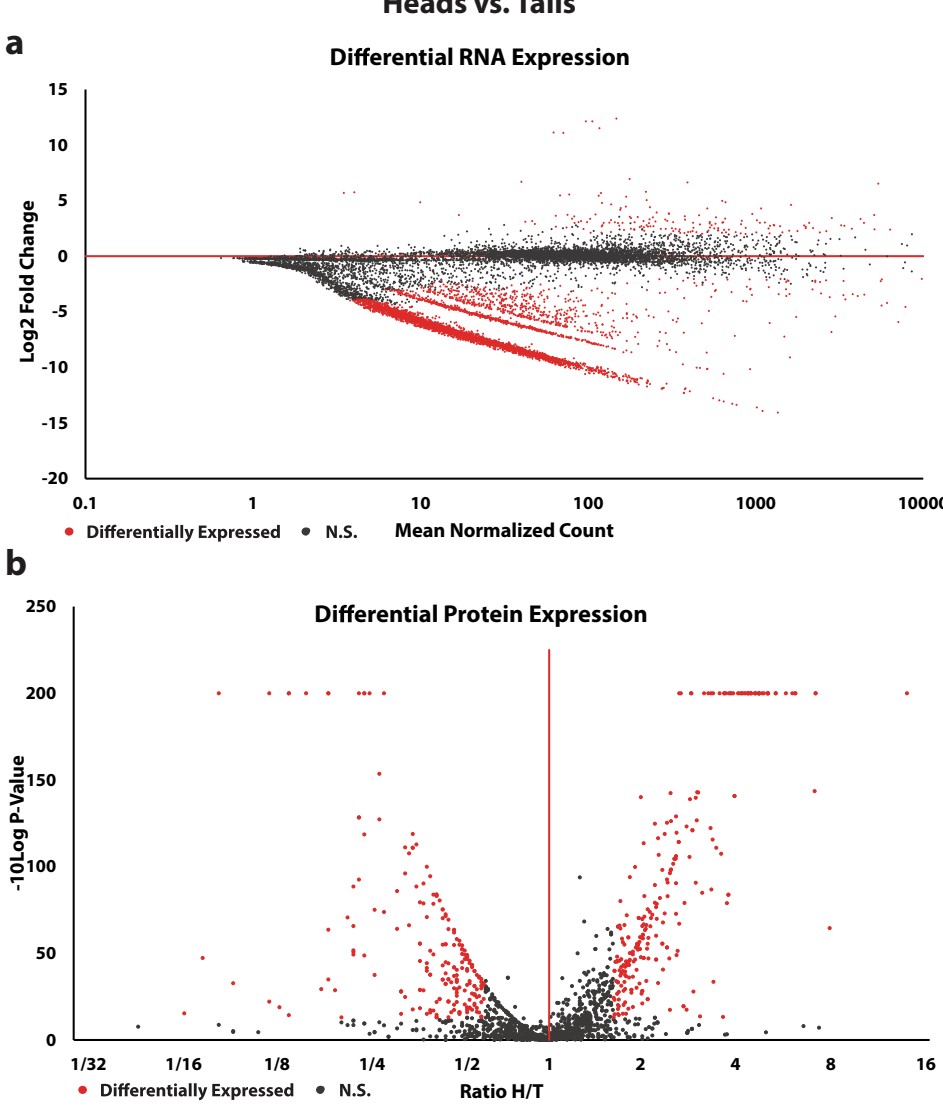

**Fig. 2 Both RNA and protein are significantly differentially expressed in cercarial heads and tails. a** shows the MA plot of heads compared to tails, with tails being the reference. All red dots (Differentially Expressed) are significantly upregulated transcripts with ≥2.0 log2 fold change and have an adjusted *p*-value of ≤ 0.01. All black dots (N.S.) are outside these thresholds. **b** shows a volcano plot of differentially expressed proteins heads and tails by the ratio (Head/Tail). All red dots (Differentially Expressed) are proteins with a ratio of ≥1.6 and a significance ≥13.0 −10 log *p*-value. All black dots (N.S.) fall outside of these thresholds. All supporting data is contained in Supplementary Data 1. Replication $n = 3$ independent biological replicates.

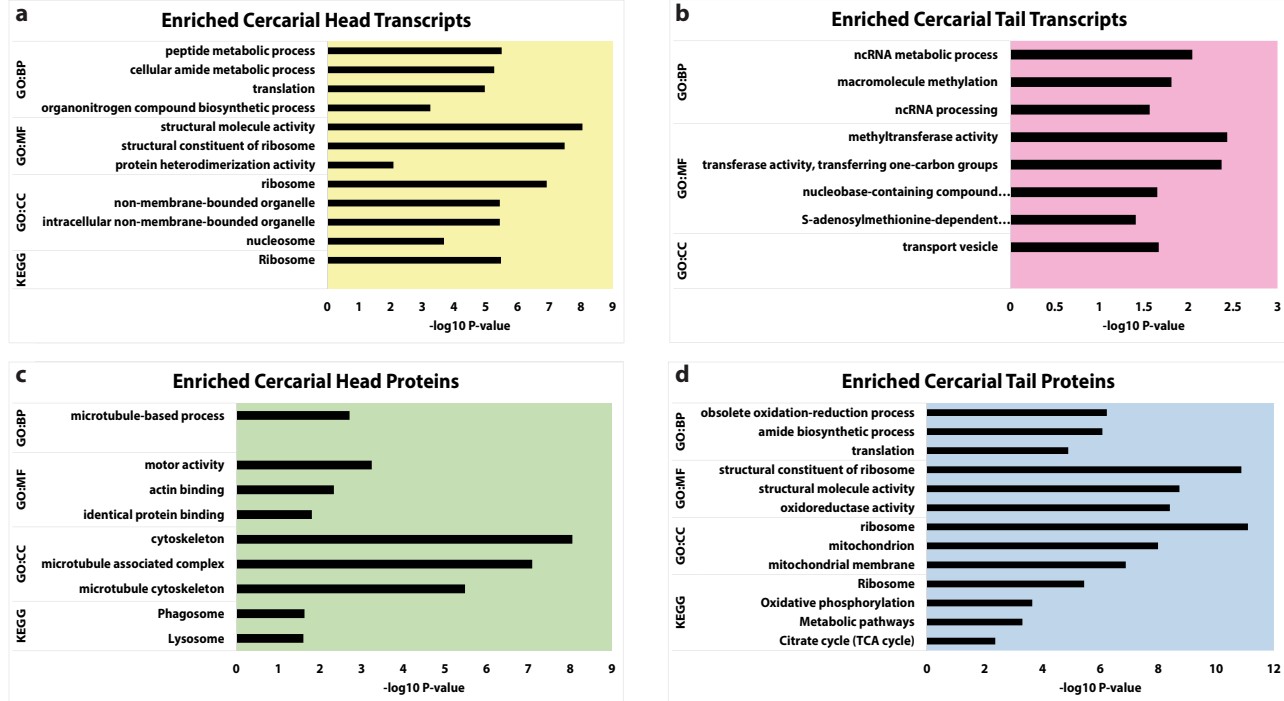

**Fig. 3 GO analysis of overexpressed transcripts and proteins from cercarial heads and tails.** Cercarial Heads have significant enrichment for ribosomal transcripts and structural proteins. Cercarial tails have significant enrichment of ribosomal and mitochondrial proteins. All data shown represent enriched categories of differentially expressed genes via GO analysis. Panels **a** and **b** show enriched transcript categories in cercarial heads and tails. **c**, **d** show enriched protein categories from cercarial heads and tails. All supporting data is contained in Supplementary Data 3.

negatives, a weighted correction method, gSCS, based on the overlap of GO categories, is employed[20]. The input proteins for GO enrichment and reduction analysis were separated into two classes: those that are unique only to heads or only to tails (Supplementary Fig. 1 and Supplementary Data 2) and those that are differentially expressed in heads and tails (Figs. 3 and 4, Supplementary Data 3). Unique proteins were analyzed separately from differentially expressed proteins because the unique proteins are removed from the label-free quantitation during retention matching of protein spectra.

We did not identify any unique enriched or reduced transcript classes or any unique reduced protein classes defined by gene ontology for cercarial heads. We Identified unique enriched cercarial head protein groups associated with the spliceosome, such as LSm3 *Smp_078640*, U1a/U2b *Smp_069870.1*, and SF3A3 *Smp_003630*) (Supplementary Fig. 1 and Supplementary Data 2). We also identified unique reduced protein groups in cercarial heads related to metabolic pathways. We found multiple enriched GO categories of unique proteins in cercarial tails, including translational and mitochondrial processes (MRPS25 *Smp_066620*, MRPS5 *Smp_332830*, and MRPL47 *Smp_102280*) (Supplementary Fig. 1 and Supplementary Data 2). The enrichment of translation and mitochondrial proteins supports previous findings showing upregulation of metabolic activity (Supplementary Fig. 1 and Data 2). Notably, heads and tails show little overlap in transcript or protein GO function. Cercarial heads lack proteins related to metabolic pathways, and cercarial tails are enriched for proteins related to metabolism, supporting the observed biology of the tail functioning as transient motility. In contrast, the more quiescent head progresses through development.

Using GO enrichment and reduction analysis, we compared differentially expressed transcripts and proteins within each macrostructure (Fig. 3, Supplementary Data 3). Head transcripts were enriched for several processes, including protein production, ribosomal biogenesis, and chromatin (*RPL27e Smp_063350*, *RPS23 Smp_074470*, *MRPS9 Smp_333040*, and *H3F3C Smp_082240*) (Fig. 3a, Supplementary Data 3), and tail transcripts were enriched for noncoding RNA processing and chromatin marking genes (*METTL1 Smp_130610*, *MARS Smp_040770*, *NAT10 Smp_144490*, and *DNMT2 Smp_334230*) (Fig. 3b, Supplementary Data 3).

Cercarial heads were enriched for structural, lysosomal, and phagosomal proteins (α-Tubulin *Smp_090120*, Paramyosin *Smp_021920*, *Smp_210500* Cathepsin L3, HEXB *Smp_053900*) (Fig. 3c, Supplementary Data 3). In contrast, cercarial tails were significantly enriched for translational, ribosomal, stress response, and mitochondrial proteins (Fig. 3d, Supplementary Data 3). These include a large population of ribosomal components (RPL11/12 *Smp_012750*, RPL24 *Smp_001830*, RPL27a *Smp_325920*, RPS9 *Smp_180000*, RPS13 *Smp_096750*, RPS15 *Smp_307550*, and RPS19 *Smp_174950*)[21–32]. The ribosomal component genes described above are involved in ribosomal biogenesis or translational upregulation[21–32].

We next analyzed upregulated proteins from three-hour, two-day, and five-day schistosomes using data published by Sotillo et al. 2015[19]. Developing schistosomula increase translation rates over multiple days after transformation[13,33]. We could not process the data using our analysis pipeline because the raw data is not publicly available. Over-expression was compared to zero-hour schistosomula. Zero-hour schistosomula are almost equivalent to cercarial heads. The GO enrichment analysis of proteins upregulated in the three time points shows enrichment of protein classes related to translation, ribosomes, and ribosomal components (Fig. 4, Supplementary Data 4). Ribosomal protein RPS13 was overexpressed in all three time points (Supplementary Data 4[19]).

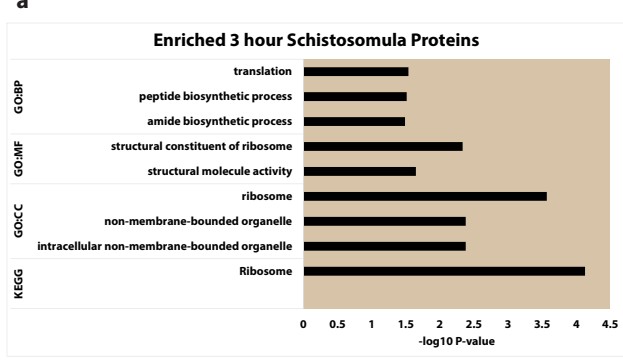

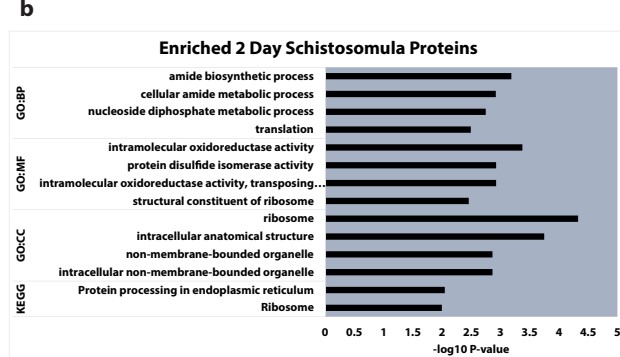

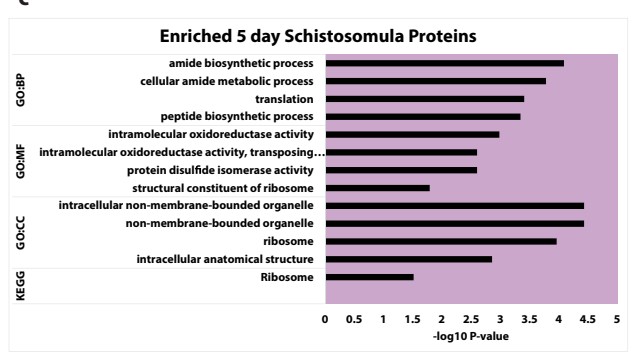

**Fig. 4 GO analysis of enriched proteins in developing schistosomula.** Schistosomula are significantly enriched for ribosomal component proteins at all three developmental time points analyzed. Over-expressed proteins from 3 h, 2 day, and 5 day schistosomula were re-analyzed from publicly available data published by Sotillo et al. 2015[19]. **a** represents enriched GO categories for overexpressed proteins in 3 h schistosomula, **b** represents the same for 2 day schistosomula, and **c** represents the same for 5 day schistosomula. All supporting GO analysis data is contained in Supplementary Data 4.

Cercarial heads have few reduced transcript or protein GO classes. The reduced GO status indicates a lack in the expected number of transcripts that fit into a given category based on the size of that category and the size of the input data set. Cercarial heads lacked membrane component transcripts and proteins (Fig. 5a, c, Supplementary Data 3). In cercarial tails, GO analysis highlighted a significant reduction of transcripts involved in translation and peptide biosynthesis. (Fig. 5b, Supplementary Data 3). At the protein level, tails have reduced proteins related to the nucleus and membrane components (Fig. 5d, Supplementary Data 3). Still, they are enriched for ribosomal and translational proteins, although transcripts are significantly lacking in these same categories. Overall, the GO analysis data show that transcript classes in heads or tails unexpectedly do not predict the protein populations.

**Transcripts in cercarial heads and tails are not predictive of translation products.** We further explore the observation that transcripts in heads or tails do not predict protein classes by testing whether the available transcripts and observed proteins aligned in these populations. We compared transcripts and proteins in heads and tails using a Venn diagram. We found that 205 overexpressed head proteins did not have corresponding transcripts, and 152 over-expressed head transcripts did not have corresponding proteins (Fig. 6a). We found that tails have 4235 over-expressed transcripts, for which we were surprisingly unable to identify any corresponding proteins (Fig. 6a). Most surprising was the intersection of over-expressed head proteins and over-expressed tail transcripts. We found 39 proteins that are over-expressed in heads that intersect with transcripts over-expressed in tails. In contrast, we observed only 11 head proteins that intersected with head transcripts. Cercarial tails have the most

robust relationship between differentially expressed transcripts and proteins (Fig. 6a). We identified 71 transcripts that were both overexpressed at the transcript and protein levels in tails. These findings suggest a weak correlation between protein and transcript levels in cercariae.

To assess the relationship between these two processes more quantitatively, we performed a correlation analysis on the transcriptome and proteome of the cercarial head and tail. We were able to quantify and match 2,254 proteins with respective transcripts from our RNA-seq datasets. We then compared 2,254 matched and quantified transcripts and proteins by their respective normalized abundance measures (Supplementary Data 1). Our analysis shows a limited correlation in cercarial head transcripts and proteins with an $R^2 = 0.0605$ and a Spearman correlation of 0.2463 (Fig. 6b). Cercarial tail transcripts and proteins are more highly correlated with an $R^2 = 0.1564$ and a Spearman correlation of 0.3756 (Fig. 6c). We have previously shown global differences in translational control and expect more specific regulation that represses genes in heads and tails[13]. Given our previous findings, the relationship between over-expressed transcripts and proteins (Fig. 6a), and the lack of correlation between transcript and protein levels (Fig. 6b, c), it is likely that the mismatch is in part due to the repression of large subsets of transcripts within heads and tails. To further explore these observations, we then analyzed the proteome and transcriptome for putatively repressed genes. We verified the RNA-seq abundance of 5 representative genes in cercarial tails using digital droplet PCR. The spearman correlation of the genes was 0.8, the analyzed genes represented high, moderate, and low abundance transcripts by normalized gene counts (Supplementary Fig. 2). We did not observe amplification in our cercarial head samples and could not repeat the tests given material limitations.

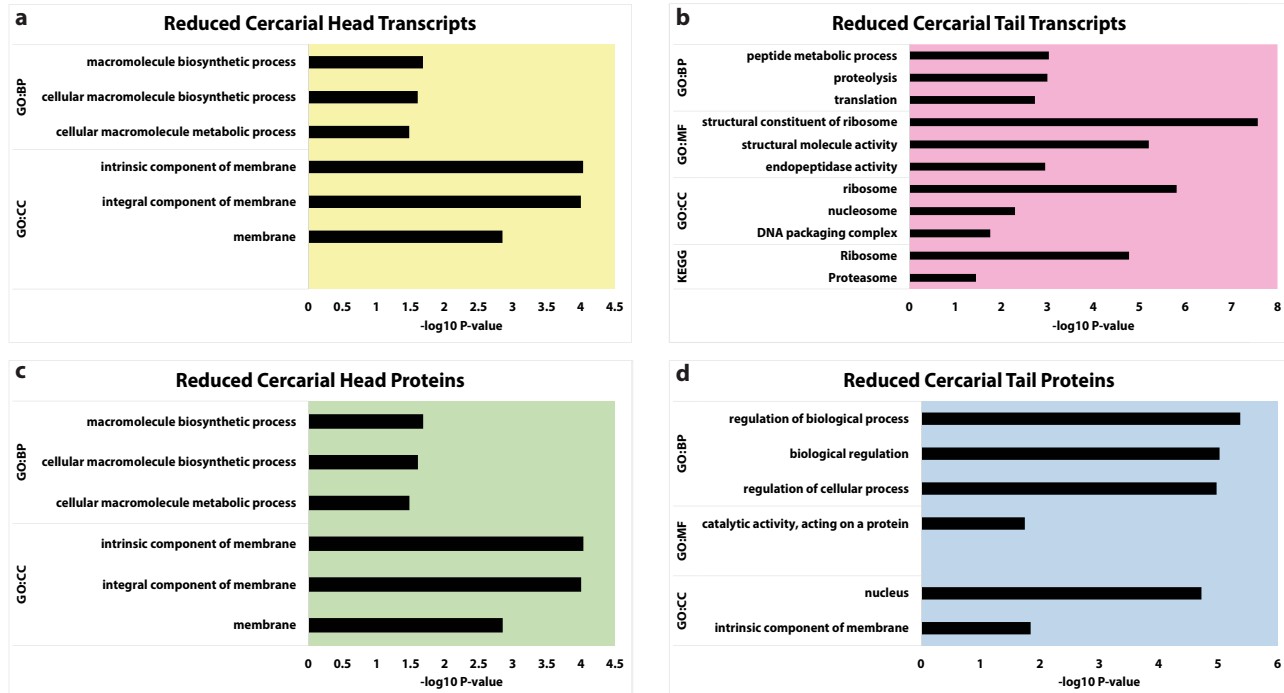

**Fig. 5 GO analysis of reduced transcripts and proteins from cercarial heads and tails.** Cercarial Tails have a significant reduction of transcripts related to ribosomal genes and translational maintenance, while cercarial heads show a reduction of both transcript and protein for membrane component genes. All data shown represent reduced categories of differentially expressed genes via GO analysis. **a**, **b** shows a reduction of transcript categories for cercarial heads and tails, respectively. **c**, **d** shows a reduction of protein categories from cercarial heads and tails, respectively. All supporting data is contained in Supplementary Data 3.

## Discussion

Previous -omic analysis of whole infectious cercariae showed upregulation of metabolic genes involved in glycolysis, including NADH dehydrogenases and proteases involved in host invasion, including cercarial elastases and serine proteases[1–8]. This work supports these previous findings while also elucidating differences in translational control mechanisms and the lack of correlation between transcription and translation[13]. Cercarial heads and tails vary significantly in overall translation rates, with heads having limited translation compared to tails[13]. This work expands on these previous observations. First, using multiple -omics approaches, we show that transcripts and proteins found in the two cercarial macrostructures are meaningfully different and correspond to the macrostructures' biological functions, quiescent storage, and structural components in heads and metabolic genes to maintain motility in tails. Second, we offer insight into protein production regulation by comparing the cercarial head and cercarial tails transcript and protein groups. Finally, we show that transcript levels and protein levels do not have a meaningful correlation in the cercarial life stage.

Cercarial heads and tails differ in structure and function. The mostly quiescent cercarial head attaches to the host, produces enzymes to facilitate host invasion, and progresses through development to an adult worm after host infection. The cercarial tail is responsible for motility, mechanically assists with host invasion, and is lost upon invasion of the host. We used RNA-seq and label-free mass spectrometry analysis to explore the molecular and regulatory differences in these structures. We identified populations of proteins and transcripts unique to the cercarial heads and tails (Fig. 1). The cercarial tail contains 7,014 unique transcripts, and the cercarial head has 207 unique transcripts (Fig. 1a, Supplementary Data 1). The unique proteins show an inverse storage pattern compared to unique transcripts, with the cercarial head containing 136 unique proteins, while the cercarial

tail has 43 unique proteins (Supplementary Data 1). We see a clear difference in the storage of unique protein and transcripts, which is somewhat unexpected given the increased translation rate in cercarial tails compared to cercarial heads. We identified 164 upregulated transcripts in cercarial heads and 4,347 upregulated transcripts in cercarial tails (Fig. 2a, Supplementary Data 1). The role of this large population of stored transcripts in cercarial tails is unclear. Increasing evidence indicates functional roles for mRNA products outside of their coding potential and UTR regulatory regions. For example, *P53* mRNA has been shown to have an auto-regulatory function on its translation, and *HIST1C* mRNA can negatively regulate telomere length [55,56]. Developmental regulator *OSK* controls embryonic patterning when translated into protein, while the mRNA independently controls karyosome formation[34]. Long noncoding RNAs are differentially regulated throughout schistosome development and could be playing functional roles in the cercarial head and tail[35,36]. The proteomic differential expression analysis reveals upregulation of 255 and 208 proteins for cercarial heads and tails, respectively (Fig. 2b, Supplementary Data 1). The abundance of unique and differentially expressed transcripts and proteins across cercarial heads and tails supports our previous assertion that the head and tail need to be treated as two distinct functional macrostructures. We were also able to affirm this with GO enrichment that identified previously unreported gene groups at both the transcriptome and proteome levels. Cercarial tails are enriched for ribosomal proteins that increase translational rates, while cercarial heads are enriched for lysosomal and phagosomal proteins related to autophagy (Fig. 3c, d, Supplementary Data 3).

Cercarial heads are enriched for ribosomal components and translation-related transcripts (Fig. 3a, Supplementary Data 3) while generally repressed for translation. These transcripts are likely stored for later translation after host invasion. Cercarial head proteins are enriched for mRNA processing, proteolysis, and

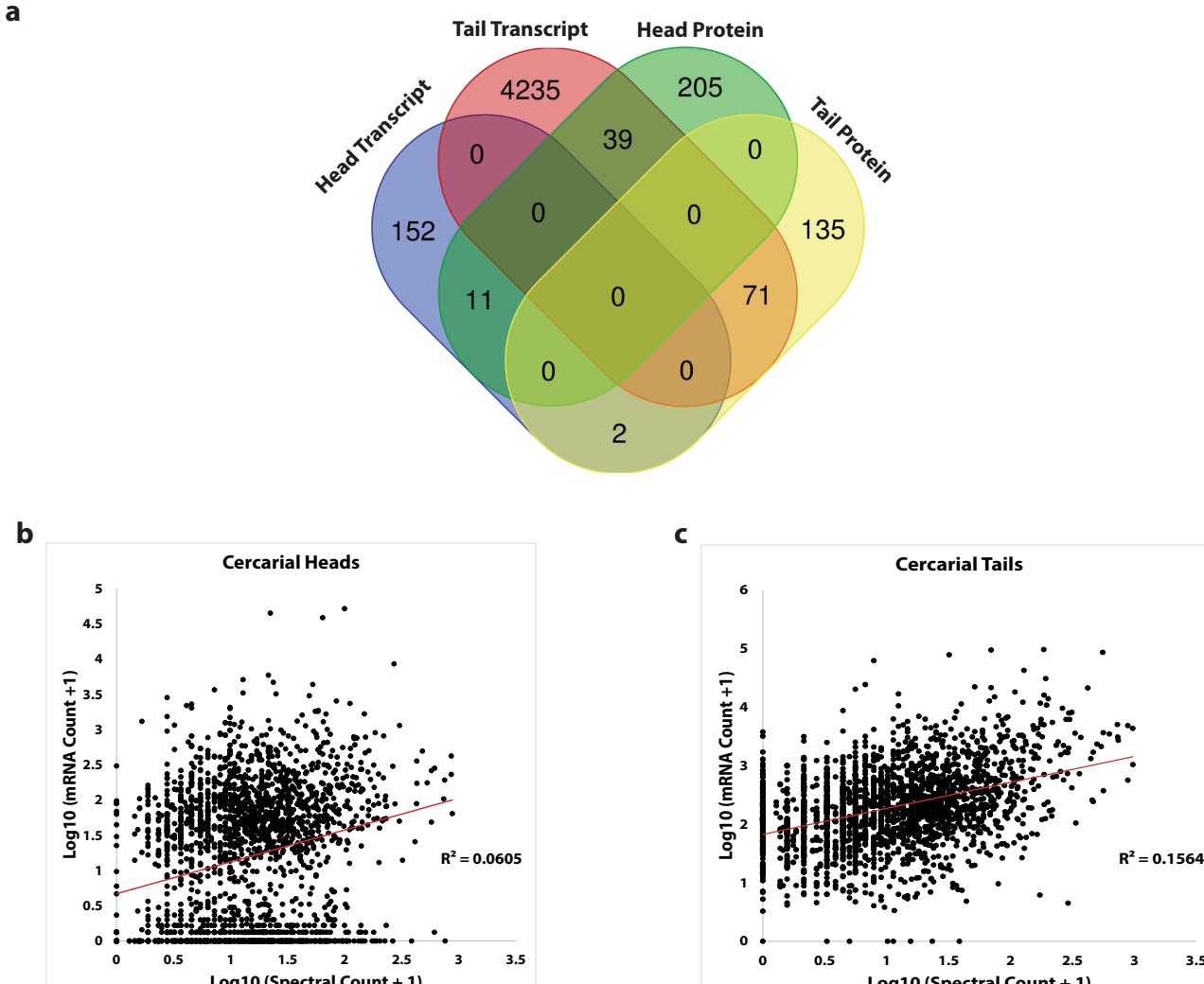

**Fig. 6 Translation and transcription are not highly correlated in cercarial heads and tails. a** shows the intersection of differentially expressed transcripts and proteins in cercarial heads and tails. **b**, **c** show a scatter plot of normalized abundance from transcript along the X-axis and normalized abundance from protein along the Y-axis for heads and tails individually. The $R^2 = 0.0605$ and $R^2 = 0.1564$ with red linear fit lines show little positive correlation between the transcript and protein abundance in cercarial heads (**b**) and cercarial tails (**c**), respectively All supporting data is contained in Supplementary Data 1. Replication $n = 3$ independent biological replicates.

cytoskeleton-related genes (Fig. 3c, Supplementary Fig. 1, Supplementary Data 2, and Supplementary Data 3). The mRNA-related proteins in cercarial heads include LSm3, U1a/U2b, and SF3A3. The storage of these mRNA-related proteins is interesting, given the stalled state of RNA production in cercariae[14,37–39]. The mRNA processing proteins may be stored for the early burst of transcription, which begins shortly after invading the definitive host[14]. These results support Roquis et al. 2015, showing that cercariae do not produce new nucleic acids but are primed for a massive burst of transcription after host invasion[14]. Furthermore, cercarial tails are enriched for differentially expressed proteins related to translation and mitochondrial maintenance MRPS25, MRPS5, and MRPL47 (Fig. 3d, Supplementary Fig. 1, Supplementary Data 2, and Supplementary Data 3). Together we see that the identification of metabolic genes from tails and proteolytic genes from heads using whole cercarial analysis only gives a shallow look at the functions of either structure[9]. The separation and individual analysis of heads and tails give insight into mechanisms of translational control and development. We find that the cercarial head is enriched with proteins that can assist in remodeling after host invasion. We can now report that the

cercarial tail is specifically storing ribosomal components and a large population of unused transcripts that had not been previously identified.

Previously, we demonstrated that cercariae have significantly different translational regulation profiles for the heads and tails[13]. Our analysis of the transcriptome and proteome suggests likely mechanisms of translational regulation in heads and tails. We observed a large population of proteins related to ribosomal biogenesis, translational initiation, and ribosomal maintenance in cercarial tails (Fig. 3d, Supplementary Data 3). These include RPL11/12, RPL24, RPL27a, RPS9, RPS13, RPS15, and RPS19[21–32]. More specifically, the ribosomal proteins, RPS19, RPL23, RPL27a, RPS13, RPS15, and RPL24, are enriched in translationally efficient polysomes compared to inefficient monosomes[40]. The opposing GO enrichment of ribosomal component proteins and GO reduction of ribosomal component transcripts is striking in the cercarial tail. The cercarial tail stores significant populations of ribosomal proteins but lacks transcripts for ribosomal proteins. (Figs. 3b and 5b, Supplementary Data 3). On the other hand, cercarial heads are enriched for ribosomal transcripts (Fig. 3a, Supplementary Data 3). Together these

**Table 1 Over-expressed (O.E.) ribosomal component proteins in cercarial tails.**

| Accession | Ribosomal Component | Cercarial Tail | 3 h Schistosomula | 2 Day Schistosomula | 5 Day Schistosomula |
|---|---|---|---|---|---|
| Smp_180000 | RPS9 | O.E. | — | O.E. | — |
| Smp_096750 | RPS13 | O.E. | O.E. | O.E. | O.E. |
| Smp_074780 | RPS18 | O.E. | O.E. | — | — |
| Smp_174950 | RPS19 | O.E. | O.E. | O.E. | — |
| Smp_012750 | RPL12 | O.E. | — | O.E. | O.E. |
| Smp_022640 | RPL13 | O.E. | O.E. | — | O.E. |
| Smp_007900 | RPL 23 | O.E. | — | — | — |
| Smp_325920 | RPL27a | O.E. | — | — | — |
| Smp_307550 | RPS15 | O.E. | — | — | — |
| Smp_001830 | RPL24 | O.E. | — | — | — |
| Smp_018990 | RPL9 | O.E. | ID | ID | ID |
| Smp_032260 | RPL15 | O.E. | — | — | — |
| Smp_038510 | RPL6 | O.E. | — | — | — |
| Smp_101450 | RPL7 | O.E. | — | — | — |
| Smp_024850 | RPL17 | O.E. | — | — | — |
| Smp_090230 | RPL13a | O.E. | ID | ID | ID |
| Smp_013470 | RPS2 | O.E. | ID | ID | ID |
| Smp_210310 | RPL7a | O.E. | — | — | — |
| Smp_175740 | RPL14 | O.E. | — | — | — |

Overexpressed (O.E.) ribosomal component proteins in the cercarial tail were compared to the dataset of Sotillo et al. 2015[19]. (O.E.) denotes over-expression, (—) denotes no additional information, (ID) denotes verification of protein without over-expression. All supporting data is contained in Supplementary Data 1 and Sotillo et al. 2015[19].

results suggest that cercariae have been primed with ribosomal proteins in the tails and have stored dormant transcripts for ribosomal components in the heads. This lack of multiple important ribosomal components could lead to global levels of translational repression[24,27,31].

We next verified the correlation of the ribosomal component proteins with increases in translational rates during development in schistosomula using publicly available proteomics data from Sotillo et al. 2015[19]. Ribosomal component proteins, (RPS9 *Smp*_180000, RPS13 *Smp*_096750, RPS18 *Smp*_074780, RPS19 *Smp*_174950, RPL12 *Smp*_012750, and RPL13 *Smp*_022640) are over-expressed in both cercarial tails and developing schistosomula (Table 1, Supplementary Data 1, and Supplementary Data 4). All six components are over-expressed in the translationally active cercarial tail and at least two other time points in schistosomula between three hours and five days of development (Table 1). RPS2 *Smp*_013470, RPL9 *Smp*_018990, and RPL13a *Smp*_090230 are over-expressed in cercarial tails but were not over-expressed in developing schistosomula, but their identification was validated (Table 1, Supplementary Data 1, and Supplementary Data 4). The cercarial tail and developing schistosomula are all more translationally active than the cercarial head[13,33,41]. Interestingly elongation factors were not overexpressed in schistosomula until day two and maintained overexpression at day five compared to zero-hour. We did not identify differential protein expression of translational initiation or elongation factors in our comparison of cercarial heads and cercarial tails (Supplementary Data 1).

We then explored the predictive quality of transcript on protein levels in cercariae using a matched and normalized correlation plot, as well as a Venn diagram showing overlapping areas of overexpressed genes (Fig. 6). We found an apparent lack of overlap between proteins and transcripts via the Venn diagram mapping of both unique and differentially expressed transcripts in heads (2.2%) total and tails (41.5%) total (Fig. 6a). Next, we analyzed the overall correlation of transcript and protein abundance using normalized counts and normalized weighted spectral counts. We found a weak positive correlation $R^2 = 0.0605$ in cercarial heads and cercarial tails $R^2 = 0.1564$ (Fig. 6b, c) between transcript and protein abundance. These values represent a weak

correlation compared to multiple studies performed on human tissues[42–44]. An analysis of 29 tissues in humans showed that most proteome and transcriptomes have a ~0.50 $R^2$ correlation[44]. The Spearman ranked correlation, 0.2463 in cercarial heads and 0.3756 in cercarial tails, reveals some predictive power though the relationships are well below the direct correlation values in human tissues[44]. Therefore, the predictive power of transcripts is low in cercariae. Given the pattern of low correlation in other systems and tissues, it is likely that this correlation is low in other developmental stages.

We have demonstrated that cercarial heads and tails are unique macrostructures that serve different functions and prepare in different ways for those roles. The cercarial head is primed with mRNA processing transcripts, motor proteins, and autophagy-related proteins to allow for the large burst of transcription and remodeling after transformation (Fig. 3a, c, Supplementary Data 1). The cercarial tail is enriched with ribosomal proteins and metabolic proteins to maintain motility and assist in honing into a potential host (Fig. 3b, d, Supplementary Data 1). Our finding suggests that the number of ribosomal proteins and ribosomal component mRNAs may be the mechanism for how translation is regulated differently between cercarial heads and tails. The storage of essential proteins allows cercarial heads to preserve energy prior to infection of the host while remaining primed to initiate the developmental changes needed after host invasion[15,45,46]. The transcriptome and proteome are not strongly correlated in cercariae. The lack of correlation could stem from posttranscriptional regulation, leading to massively different translational rates that vary by five orders of magnitude[47].

Our findings have opened new questions. What role do the large populations of stored transcripts in tails that do not appear to be translated play in this transient structure that does not progress through development? Interestingly the cercarial tail can develop after separating the head and proliferate new cells and growths in culture[48]. The signals for transformation after the loss of the tail that leads to growth remain unclear. Although the tail is not directly involved in development after transformation, it may store transcripts and proteins related to development that receive a similar signal to begin growth when separated from the head. Furthermore, we know mRNAs escape translational repression

and are translated in the head and the early schistosomula. What unique patterns and regulatory elements can be utilized for tools in schistosomes? Are these patterns of unique expression different across cell populations, not only the macrostructures? We hope to further explore the regulatory mechanisms of translation in cercariae as they develop toward adulthood, particularly after host invasion in early schistosomula. We propose two likely mechanisms for translational regulation in cercariae: ribosomal quantity and ribosomal heterogeneity. Ribosomal quantity means an increased overall abundance of actively translating ribosomes are available; thus, translational rates are increased. Ribosomal heterogeneity proposes a different model that does not require a difference in the number of ribosomes but differences in the type of available ribosomes. These differences can include different ratios of ribosomal component proteins, differences in rRNA, and differences in post-translational modifications. We identified seven overexpressed ribosomal proteins found in higher ratios in actively translating polysomes[40]. These targets of nascent translation in schistosomula can give insight into early founder proteins and minimally required translation for maintenance of the organism.

## Methods

**Parasite collection**. Biomphalaria glabrata snails infected with Schistosoma mansoni (NMRI strain) were obtained from the Biomedical Research Institute (BRI; Rockville, MD). Cercariae were shed from infected B. glabrata snails were kept in total darkness for 48 h before exposure to intense white light for 1.5 h to release cercariae. Cercariae were collected then concentrated at $1000 \times g$ for 15 min at 4 °C[49].

**Cercarial head and tail separation**. After collection, cercariae were incubated in incomplete DMEM (Gibco) with 10% ethanol supplementation to prevent further development. Cercariae heads and tails were then separated by vortexing and passing through a 22-gauge needle. The mixed head and tail samples were washed with incomplete DMEM and segregated using an ice-cold 70% percoll gradient and centrifugation at $1000 \times g$ for 25 min at 4 °C. Head and tail fractions were washed with incomplete DMEM 3 times and visualized for contamination. Samples were then washed in 1x PBS 3 times, pelleted, and flash-frozen[13].

**RNA extraction**. RNA was extracted from ~60,0000 cercarial heads and cercarial tails per replicate using the Direct-zol RNA Miniprep Kit (Zymo Research, Irvine, CA) following their standard protocol, including on-column DNAseI digestion. RNA concentration was assessed using a Nanodrop 8000 spectrophotometer (Thermo Scientific, Waltham, MA). Library preparation and barcoding were performed using the Pico stranded library prep kit (Clontech, Mountainview, CA).

**Genomic and transcriptomic data analysis**. The S. mansoni genome sequence and annotation were downloaded from WormBase ParaSite[50,51]. The most recent version (release 14) was used for the analysis presented here.

Six RNA-Seq datasets were used for this study: three sets of in-house cercarial head dataset composed of ~60 million raw paired-end reads from cercarial heads and three sets of in-house cercarial tail dataset composed of ~80 million raw paired-end reads from cercarial tails. Data collection was performed using Illumina HiSEQ 2500 for both head and tail samples. The datasets were checked for quality using FastQC[52], and the adapters were trimmed using Trimmomatic[53]. The datasets were then aligned to the S. mansoni genome using HISAT2[54]; then, the transcripts were assembled with the S. mansoni annotation as the reference using Stringtie[55]. The transcripts were quantified using the mapping mode of Salmon[56] with the S. mansoni genome sequence and the annotation used as the reference transcripts.

Additional filtering to remove any transcripts that match any snail host transcript was performed. The three sets of uninfected B. glabrata RNA-seq datasets (accession number PRJNA602007) were quality-checked, trimmed, aligned, and assembled as described above. Three transcripts that match snail transcripts were further removed prior to the differential gene expression analysis.

To further filter the poorly characterized putative genes and pseudogenes in the S. mansoni reference annotation, the standalone version of riboPicker[57] was used on the reference transcript sequence to identify and remove 29 transcripts that match rRNAs. Multiple databases were used for the riboPicker process. SILVA rRNA database was used for large subunit (version 132), and small subunit (version 138) rRNA sequences[58]. Rfam was used for 5 S and 5.8 S subunit rRNA sequences (release 14.2)[59,60]. As the SILVA small subunit sequence data file was too large to index using riboPicker, it was broken into 6 similar-sized files for indexing, and the large, 5 S, 5.8 S, and the 6 small subunit indices were used together for the analysis.

Statistical analyses were performed with the R environment 4.02 v3.4.4: with DESeq2[18], readr[61], and tximport[62] libraries loaded. The log2 fold change shrinkage was performed with the apeglm package[63].

**Digital droplet PCR**. Absolute quantitation of representative transcripts was performed via digital droplet PCR using the Biorad QX200 system. Biorad Eva-Green Supermix was added at a 1x concentration to 20 μL reactions. Each reaction used a primer concentration of 150 μM, and reactions were performed in duplicate. Each reaction contained 10 ng input cDNA or RNA only for negative controls. No template controls were performed in single replicates. cDNA generation using Superscript kit following the standard protocol with oligodT primer. Quantitation was performed using ddpcRquant webtool[64]. No template control samples were merged for threshold analysis, and default settings were used for processing[64]. Primer sequences are in Supplementary Data 1. Thermocycler conditions: 95 C for 5 min, 95 C for 30 sec and 60 C for 1 min 40 cycles, 4 C for 5 min, 90 C for 5 min.

**Protein sample preparation**. Cercarial heads and tails, ~60,000 per replicate, were thawed on ice for 30 min in a 300 μL 2% SDS and protease inhibitor cocktail (Sigma, St. Louis, MO). Sonication was performed on all samples at 50% amplitude with a probe sonicator then vortexed; this procedure was repeated for four cycles with an intervening ice incubation between cycles. SDS detergent removal and alkalylation were performed following the FASP protocol[65].

After homogenization was complete, samples were processed using the FASP protocol and Amicon Ultra MWCO 3 K filter (Millipore, Billerica, MA). Samples were reduced and akalylated on filter with 10 mM dithiothreitol (Acros, Fair Lawn, NJ) and 25 mM iodoacetamide (Acros, Fair Lawn, NJ). Samples were then concentrated to a final volume of 40 μL in 8 M urea. The sample concentration was determined using the Bradford assay kit (Bio-Rad, Hecules, CA).

Following sample cleanup using FASP, 10 μg of total protein was aliquoted for digestion. The concentration of urea was reduced to 4 M using 50 mM Tris pH 8. Protein digestion was performed using mass spectrometry grade lysyl endopeptidase (Wako Chemicals, Richmond, VA) using a protease enzyme to substrate ratio of 1:40 and 2-hour incubation at 37 °C. Following lysyl digestion, the urea concentration was further reduced to 2 M using 50 mM Tris pH 8, and samples were digested using sequencing grade trypsin (Promega, Madison, WI) at a digestion enzyme to substrate ratio of 1:40 overnight at 37 °C. Samples were then diluted using 0.1% formic acid (Thermo Scientific, Rockford, IL) for LC–MS/MS analysis.

**Reverse phase LC–MS/MS**. Sample injections of 11 μL containing 600 ng digested peptide was loaded with blank runs intervening between each sample. The cercarial heads and tails were run in triplicate. The Orbitrap Velos Elite mass spectrometer (Thermo Electron, San Jose, CA) equipped with the Waters nanoACQUITY LC system (Waters, Taunton, MA) was used for acquisition. Peptides were desalted in a trap column (180 μm × 20 mm, packed with C18 Symmetry, 5 μm, 100 Å, Waters, Taunton, MA) and subsequently resolved in a reversed-phase column (75 μm × 250 mm nano column, packed with C18 BEH130, 1.7 μm, 130 Å (Waters, Taunton, MA). Liquid chromatography was carried out at ambient temperature at a flow rate of 300 nL/min using a gradient mixture of 0.1% formic acid in water (solvent A) and 0.1% formic acid in acetonitrile (solvent B). The gradient employed ranged from 4 to 44% solvent B over 210 min. Peptides eluting from the capillary tip were introduced into the nanospray mode with a capillary voltage of 2.4 kV. A full scan was obtained for eluted peptides in the range of 380–1800 atomic mass units, followed by twenty-five data-dependent MS/MS scans. MS/MS spectra were generated by collision-induced dissociation of the peptide ions at a normalized collision energy of 35% to create a series of b- and y-ions as major fragments. A one-hour wash was included between each sample.

**Protein identification and label-free quantitation**. All identification and quantitation were performed using Peaks X + (Bioinformatics Solutions Inc., Waterloo, ON, CA)[66–68]. De Novo sequencing from spectra was performed with a parent mass error tolerance of 15.0 ppm and a fragment mass error tolerance of 0.5 Da. Fixed and variable modifications accounted for in sequencing: Fixed-carbamidomethylation 57.02, and variable-deamidation (NQ) 0.98, oxidation (M) 15.99. The max variable PTM per peptide was set at 3. Following de novo sequencing PEAKS database search using Wormbase parasite protein annotation release 14 with cRAP contaminants database. Following PEAKS database search, PEAKS PTM search was performed with a de novo score threshold of 15, and a peptide hit threshold of 30.0 (−10logP). Following PEAKS PTM search, the SPIDER Homology match search was performed. Following SPIDER homology search ID- directed label-free quantification was performed using the aforementioned PEAKS, PEAKS PTM, and SPIDER searches. The mass error tolerance was set to 20.0 ppm, and retention time shift tolerance was set to 20.0 min. Total ion current (TIC) normalization was applied to all samples. The false discovery rate (FDR) threshold was set to 0.05. The threshold utilized for significant differential expression of proteins was a −10logP significance of ≥13.0 and a Head/Tail (H/T) ratio ≥ 1.6 and ≤0.62 for total reporting. All proteins specifically discussed require a more stringent H/T ratio of ≥2.0 and ≤0.50. All identified proteins were detected with ≥2 unique peptides and detected in ≥2 of the analyzed replicates. Unique

proteins are identified in ≥2 replicates in one macrostructure and 0 replicates in the other macrostructure.

**mRNA and protein abundance**. DESeq2 normalized counts were used for correlation mapping and translational repression analysis. PEAKS identifications were analyzed using Scaffold PerSPECtive (version 3.1.0, Proteome Software Inc., Portland, OR) to determine normalized weighted spectral counts. Normalized weighted spectral counts were utilized for correlation mapping.

**Gene ontology analysis**. Gene ontology (GO) analysis was performed using gProfiler with Wormbase parasite *S. mansoni* GO annotations[20]. The significance thresholds for enrichment or reduction were set at 0.05 and used the g:SCS algorithm for analysis[20]. All significance is reported as –log10 P-value. The gProfiler tool was utilized for both protein and mRNA GO analysis. Gene Ontology and KEGG pathway enrichment and reduction are reported.

**Correlation and Venn diagram mapping**. Global mRNA and protein abundance were plotted as log10 (normalized count +1) and log10 (normalized weighted spectral count + 1). The comparison of the proteome to the transcriptome utilized linear $R^2$ correlation and Spearman ranked correlation. The log10 counts were plotted against each other as a direct correlation scatter plot.

mRNA abundance measured by RNA-seq was plotted against mRNA abundance as measured by ddPCR using log10 of normalized count from RNA-seq and log10 of copies/μL from ddPCR quantitation. Linear $R^2$ and Spearman ranked correlation are reported. The plot was a direct correlation scatter plot.

**Statistics and reproducibility**. Statistical tests used in this study were carried out using DESeq2 (RNA-seq), PEAKSX, and Scaffold PerSPECtive as described above. All omic analysis was performed in triplicate with three biological replicates. All p-values and adjusted p-values for -omic identification and differential expression analysis are included in Supplementary Data 1 and Supplementary Data 4. GO analysis was performed using gProfiler and the g:SCS algorithm. All GO analysis p-values are reported in Supplementary Data 2-4. The comparison of normalized transcript (DESeq2 normalized counts) and protein abundance (Scaffold Per-SPECtive normalized weighted spectral counts) was performed using linear regression and Spearman ranked correlation analysis.

**Reporting summary**. Further information on research design is available in the Nature Research Reporting Summary linked to this article.

## Data availability

All data supporting the findings of this paper are contained within the paper or with the provided Supplementary Information and Supplementary Data files. A summary of source data for each figure is listed in the figure legends and the Supplementary Information file. The mass spectrometry proteomics data have been deposited to the ProteomeXchange Consortium via the PRIDE partner repository with the dataset identifier PXD026435 and https://doi.org/10.6019/PXD026435[69]. The RNA-Seq data analyzed in this study are available on the National Center for Biotechnology Information (NCBI) database under the BioProject PRJNA734345[70]. All other data are available from the corresponding author upon reasonable request.

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

## Acknowledgements

We would like to thank the entire Jolly lab for review and edits to the final manuscript. Research reported in this publication was supported by the following grants: National Institutes of Allergy and Infectious Disease of the NIH, grant R21AI137577; B. glabrata snails were provided by the NIAID Schistosomiasis Resource Center of the Biomedical Research Institute (Rockville, MD) through NIH-NIAID Contract HHSN272201700014I for distribution through BEI Resources.

## Author contributions

E.J. and J.H. were responsible for the project conceptualization. J.H. and H.C.K. were responsible for experimentation and methodology. E.J. was responsible for Project administration and Resources. J.H., E.J., and H.C.K. were responsible for manuscript writing. E.J., J.H., and H.C.K. were responsible for the review & editing of the manuscript.

## Competing interests

The authors declare no competing interests.
