## [Peer Review File · Communications Biology]

Reviewers' Comments:

Reviewer #1:

Remarks to the Author:

This is a well-conceived and executed study which contributed to an important aspect of the schistosome cell biology. The parasite is both an interesting example of the biology of parasitism and a pathogen causing wide range morbidity in humans.

Some suggestions to provide better clarity from the study is presented here-

Lines 69-70 Given the focus of the study I would like to see a more detailed account of the separation and isolation methods for the tail and head. This is important to ensure and determine purity of preparations. What number of parasites were used?

Were any reads to the snail host detected? If so how were these removed?

While probably out of the scope of this comprehensive study, WISH in situ would have been a good validation tool. This could be discussed.

The use of the terms over/under rep is a little confusing, could instead the authors consider just discussing cercarial enriched or tail enriched proteins/transcripts? Or reduced?

The lack of transcript to protein correlation is interesting. Does the lack of protein hits for unique transcripts suggest a limitation of proteomic sensitivity? Could you correlate absent proteins with low read transcripts?

While present transcripts didn't equate to proteins in some cases, was the opposite situation also detected? This later situation may be present in the head region?

Reviewer #2:

Remarks to the Author:

The manuscript from Hagerty et al. adds new transcriptomic and proteomic information to the cercarial stage of the blood fluke *Schistosoma mansoni*. The study is well designed, although I have reservations about the conclusions and novelty of the work. For instance, authors claim that cercarial heads and tails "differentially regulate translation using ribosomal component" and "utilize distinct translational repression regimes" (lines 59-61). In this regard, the only data supporting these claims are mass spectrometry-based data, which, in my opinion should be validated using other approaches. Furthermore, they perform an in silico translational repression analysis "as the one reported by Lindner et al." (1), although in that study, repression was validated using immunofluorescence experiments with wild type and genetically modified parasites (1), which is not the case in the current study. Also, the authors have already shown that cercarial tails and heads possess distinct transcriptional factors, with cercarial heads being mostly transcriptionally and translationally quiescent (2), so the novelty in this sense is compromised by previous work by the same authors.

The present study focuses on identifying the transcripts and proteins present in each part of the cercaria (head and tail). In this sense, there are previous studies that separate tails from heads using mechanical methods similar to the one used in this study that also perform proteomic and transcriptomic analyses in the newly transformed schistosomula (what would correspond to cercarial head from this study), and it is very surprising that authors have not tried to correlate or validate their results with these studies (3-4). No proteomic or transcriptomic studies have been performed in the tail, so this would be the only novel aspect of this study.

Furthermore, the proteomic analysis lacks methodological robustness (i.e. no data normalization, validation, etc.). For instance, several proteins have been identified with only 1 peptide. It is important to identify proteins with unique peptides (at least 2 unique peptides). Also, for the differential expression analysis, some proteins analysed (and shown in results and/or discussion) were identified in only one of the three replicates analysed from tails (i.e. Smp_063350). This is very surprising since that particular protein was identified and quantified at the same levels in one batch from tails than at the 3 batches from heads, so one would assume to find it in all batches if the abundance was similar. These results agree with the notion that ribosomal proteins are enriched in tails, but if detected in only 1 sample, they should be validated using other targeted approaches such as MRM or SRM.

In general, the study is well designed, although it lacks some methodological robustness (an issue

that can definitely be solved by the authors in future versions of the manuscript) and in vivo validation of the results. As it is, the manuscript is a proteomic and transcriptomic descriptive manuscript, with the translational repression analysis only performed in silico and the role for specific proteins (particularly ribosomal) not validated using further experiments.

References:

1. Lindner, S.E., Swearingen, K.E., Shears, M.J. et al. (2019). Transcriptomics and proteomics reveal two waves of translational repression during the maturation of malaria parasite sporozoites. *Nat Commun* 10, 4964. <https://doi.org/10.1038/s41467-019-12936-6>
2. Hagerty JR, Jolly ER (2019). Heads or tails? Differential translational regulation in cercarial heads and tails of schistosome worms. *PLOS ONE* 14(10): e0224358.
3. Protasio AV, Dunne DW, Berriman M (2013). Comparative Study of Transcriptome Profiles of Mechanical- and Skin-Transformed *Schistosoma mansoni* Schistosomula. *PLOS Neglected Tropical Diseases* 7(3): e2091. <https://doi.org/10.1371/journal.pntd.0002091>
4. Sotillo J, Pearson M, Becker L, Mulvenna J, Loukas A (2015). A quantitative proteomic analysis of the tegumental proteins from *Schistosoma mansoni* schistosomula reveals novel potential therapeutic targets, *International Journal for Parasitology*, Volume 45, Issue 8, 2015, Pages 505-516, <https://doi.org/10.1016/j.ijpara.2015.03.004>.

Dear Reviewers:

Thank you for your careful review, thoughtful comments and critiques of our original manuscript entitled “A Tail of Two Structures: Proteomic and Transcriptomic analysis of cercarial heads and tails of schistosome worms.” We have worked diligently to incorporate your suggestions into the revised document and consequently, we think that the product is now a better story. Point by point responses are included below.

Reviewer 1

This is a well-conceived and executed study which contributed to an important aspect of the schistosome cell biology. The parasite is both an interesting example of the biology of parasitism and a pathogen causing wide range morbidity in humans.

Some suggestions to provide better clarity from the study is presented here-

We outline below the steps we have taken to address the specific concerns raised below:

Lines 69-70 Given the focus of the study I would like to see a more detailed account of the separation and isolation methods for the tail and head. This is important to ensure and determine purity of preparations.

We have added additional detail outlining our separation methods for cercarial heads and tails (lines 52-57). We used vortexing and 22-gauge needle passages in ethanol supplemented media to remove heads from tails and stop the progression of cercariae into schistosomula. We then separated the head and tail material using a 70% percoll gradient. Finally, we visualized samples for contamination.

What number of parasites were used?

We added numbers of individuals used for RNA and protein extraction (lines 59 and 100). We used ~60,000 heads and ~60,000 tails for each sample in both RNA and protein preparations.

Were any reads to the snail host detected? If so how were these removed?

We have removed any transcripts that align to the schistosome genome from the uninfected snail host RNA-seq to the schistosome genome as previously described in ¹. (Lines 75-78)

The use of the terms over/under rep is a little confusing, could instead the authors consider just discussing cercarial enriched or tail enriched proteins/transcripts? Or reduced?

The use of over and underrepresentation was changed to enriched or reduced throughout the document for additional clarity. We appreciate this suggestion. It clarifies the comparison of differential expression and GO analysis throughout the manuscript.

The lack of transcript to protein correlation is interesting. Does the lack of protein hits for unique transcripts suggest a limitation of proteomic sensitivity?

Could you correlate absent proteins with low read transcripts?

While present transcripts didn't equate to proteins in some cases, was the opposite situation also detected? This later situation may be present in the head region?

We have removed the repression analysis from the manuscript and also the unique transcripts. They had not been filtered as previously thought and are accounted for during the LFC shrinkage analysis of DeSeq2. This is described in the document (lines 189-191). We hope to revisit this analysis when we have the tools to validate our analysis. To this end, we performed digital droplet PCR analysis to determine the overall abundance of target genes for cercarial tails. We hoped to complete the same analysis for cercarial heads, but material limitations prevented us from repeating the analysis. We report the findings as Supplementary Figure 2 with the primer sequences reported in Supplementary Data 1 (lines 297-300). We found a spearman correlation between our normalized counts for RNA-seq and normalized abundance by ddPCR of 0.8, giving us increased confidence in our abundance data from the RNA-seq analysis.

Reviewer 2

The manuscript from Hagerty et al. adds new transcriptomic and proteomic information to the cercarial stage of the blood fluke *Schistosoma mansoni*. The study is well designed, although I have reservations about the conclusions and novelty of the work.

We believe we have appropriately revised the manuscript to address the concerns raised by reviewer 2. We outline below the steps we have taken to address your concerns.

For instance, authors claim that cercarial heads and tails “differentially regulate translation using ribosomal component” and “utilize distinct translational repression regimes” (lines 59-61). In this regard, the only data supporting these claims are mass spectrometry-based data, which, in my opinion should be validated using other approaches. Furthermore, they perform an in silico translational repression analysis “as the one reported by Lindner et al.” (1), although in that study, repression was validated using immunofluorescence experiments with wild type and genetically modified parasites (1), which is not the case in the current study.

Given the current technical and biological limitations within cercariae, we cannot perform functional genomic techniques to confirm the role of ribosomal components or validate translational repression. Given our inability to validate the repression analysis at this time, we have decided to remove that analysis, and we believe this does not harm the overall impact or major findings of the manuscript.

Also, the authors have already shown that cercarial tails and heads possess distinct transcriptional factors, with cercarial heads being mostly transcriptionally and translationally quiescent (2), so the novelty in this sense is compromised by previous work by the same authors.

Our group has not shown distinct transcriptional factors, or that transcription does not occur in the cercarial stage. The Grunau group has demonstrated that cercariae are transcriptionally quiescent and have specific epigenetic markers throughout the genome, indicating many primed genes that begin a rapidly transcribing after transformation schistosomula ². No difference was observed in transcriptional regulation in cercarial heads and tails because both were completely suppressed. Our previous work reported that global translation rates are significantly lower in cercarial heads compared to cercarial tails. Still, we had no evidence of the distinct stored transcripts and proteins that could account for potential mechanisms of differential translational regulation ³. Our work in this current manuscript begins to elucidate likely mechanisms for this regulation we observed based on stored translational machinery that differs between the two macrostructures. We also provide further evidence showing that the cercarial stage should be treated as two separate macrostructures for omic analysis, given they have significantly different transcriptomes and proteomes.

The present study focuses on identifying the transcripts and proteins present in each part of the cercaria (head and tail). In this sense, there are previous studies that separate tails from heads using mechanical methods similar to the one used in this study that also perform proteomic and transcriptomic analyses in the newly transformed schistosomula (what would correspond to cercarial head from this study). It is very surprising that authors have not tried to correlate or validate their results with these studies (3-4). No proteomic or transcriptomic studies have been performed in the tail, so this would be the only novel aspect of this study.

The suggestions for comparison to Sotillo et al. for proteomics and Protasio et al. for transcriptomics were appreciated ^{4,5}. We have utilized a modified mechanical transformation protocol that uses ethanol during the mechanical separation steps and is followed by immediate snap-freezing to stop progression into the schistosomula stage. This distinction is most important for the transcriptome work given the high levels of transcription observed after transformation. Translation is slower to begin in early schistosomula, but protein degradation can begin quickly given 218/240 dysregulated proteins are downregulated in 3-hour schistosomula compared to uncultured schistosomula ^{4,5}.

We were able to compare overall protein identification with the Sotillo et al. data set, and we found that 684 of our 2401 proteins were validated [4]. The validation is reported within Supplementary Data 4 (lines 193-194). We reanalyzed the over-expression analysis performed by Sotillo et al. using gProfiler GO analysis and found a similar pattern of ribosomal component enrichment observed in the cercarial stage. We report this GO analysis as Figure 4. And Supplementary Data 4 ⁴. (lines 255-261). We also compare the over-expressed and identification verified ribosomal components in Table 1 and the discussion (lines 371-383).

We are unable to perform a reanalysis of the raw data because it is not publicly available ^{4,5}. The data also did not represent a major finding of the initial publication, and the unbound protein fraction that contains the differential expression data we reanalyzed in not examined thoroughly. It is mentioned as validation of select tegumental proteins. The work from Protasio et al. does not provide a comparable sample set ^{4,5}.

The transcriptomic analysis performed in that study utilized schistosomula that had been cultured for 24 hours after transformation. Given the large burst of transcription after transformation, the expression profiles of cercarial heads and 24-hour schistosomula are very different². The difference has been shown in comparisons of whole cercariae and 3-hour schistosomula, and 3 and 24-hour schistosomula, with 1518 and 1028 differentially expressed genes between the compared time points, respectively⁶. Given the aforementioned differences in both prior studies, they do not significantly diminish the novelty of this work. We have provided a large and high confidence increase in protein identification and differential expression analysis for the cercarial stage.

Furthermore, the proteomic analysis lacks methodological robustness (i.e. no data normalization, validation, etc.).

We appreciate the comments regarding robustness of the methodology and have incorporated the suggested changes. We had performed normalization of the label-free differential expression, but it was not clearly noted in the methods. We have added information on normalization performed in PEAKSX+ using total ion current normalization (lines 126-127).

For instance, several proteins have been identified with only 1 peptide. It is important to identify proteins with unique peptides (at least 2 unique peptides). Also, for the differential expression analysis, some proteins analyzed (and shown in results and/or discussion) were identified in only one of the three replicates analyzed from tails (i.e. Smp_063350). This is very surprising since that particular protein was identified and quantified at the same levels in one batch from tails than at the 3 batches from heads, so one would assume to find it in all batches if the abundance was similar.

The protein identification was altered to utilize a minimum of 2 unique peptides for any identified protein and detection in a minimum of 2 of the 3 replicates (lines 130-131). We also defined unique proteins more stringently. Unique proteins are now defined as only those proteins identified with ≥ 2 unique peptides and detected in ≥ 2 samples in one macrostructure and with 0 detections in the other macrostructure (lines 148, 194-199). The changes we have adopted match or exceed the criteria used in⁴, and also more recent proteomic publications in *S. mansoni* and other helminths^{7,8}. These changes to our thresholding have not changed the patterns we see in the data but have increased our confidence in the strength of the data.

We reduced the number of identified proteins from 3,108 to 2,401 (lines 180-181), a large proportion of these proteins were unannotated hypothetical genes. The number of unique proteins went from 601 to 555 (lines 181-182). The removal of these less stringently identified proteins did not change the enriched categories of the GO analysis but did facilitate the removal of 2 named genes from the text, RPL36, and SARS (lines 215-216). No other genes mentioned in the text were affected by the more stringent threshold. The gene RPL27e Smp_063350 is mentioned in the text as a transcript that is upregulated in cercarial heads, and that has not changed given the transcript was identified in all three replicates of both the head and the tail samples. The RPL27e protein was removed from our differential protein expression (Supplemental Data 1) because of the lack of identification in multiple samples. The RPL27e protein was not discussed in the results or methods sections. The differentially expressed proteins went from 530 to 463, the pattern of enriched ribosomal genes in cercarial tails was not changed. **These results agree with the notion that ribosomal proteins are enriched in tails, but if detected in only 1 sample, they should be validated using other targeted approaches such as MRM or SRM.**

We have increased the stringency of detection as recommended and also validated candidates using publicly available data from Sotillo et al. ⁴

- 1 Kim, H. C., Khalil, A. M. & Jolly, E. R. LncRNAs in molluscan and mammalian stages of parasitic schistosomes are developmentally-regulated and coordinately expressed with protein-coding genes. *RNA Biol* **17**, 805-815, doi:10.1080/15476286.2020.1729594 (2020).
- 2 Roquis, D. *et al.* The Epigenome of *Schistosoma mansoni* Provides Insight about How Cercariae Poise Transcription until Infection. *PLoS Negl Trop Dis* **9**, e0003853, doi:10.1371/journal.pntd.0003853 (2015).
- 3 Hagerty, J. R. & Jolly, E. R. Heads or tails? Differential translational regulation in cercarial heads and tails of schistosome worms. *PloS one* **14**, e0224358, doi:10.1371/journal.pone.0224358 (2019).
- 4 Sotillo, J., Pearson, M., Becker, L., Mulvenna, J. & Loukas, A. A quantitative proteomic analysis of the tegumental proteins from *Schistosoma mansoni* schistosomula reveals novel potential therapeutic targets. *Int J Parasitol* **45**, 505-516, doi:10.1016/j.ijpara.2015.03.004 (2015).
- 5 Protasio, A. V., Dunne, D. W. & Berriman, M. Comparative study of transcriptome profiles of mechanical- and skin-transformed *Schistosoma mansoni* schistosomula. *PLoS Negl Trop Dis* **7**, e2091, doi:10.1371/journal.pntd.0002091 (2013).
- 6 Protasio, A. V. *et al.* A systematically improved high quality genome and transcriptome of the human blood fluke *Schistosoma mansoni*. *PLoS Negl Trop Dis* **6**, e1455, doi:10.1371/journal.pntd.0001455 (2012).
- 7 Kifle, D. W. *et al.* Proteomic analysis of two populations of *Schistosoma mansoni*-derived extracellular vesicles: 15k pellet and 120k pellet vesicles. *Mol Biochem Parasitol* **236**, 111264, doi:10.1016/j.molbiopara.2020.111264 (2020).
- 8 Logan, J. *et al.* Comprehensive analysis of the secreted proteome of adult *Necator americanus* hookworms. *PLoS Negl Trop Dis* **14**, e0008237, doi:10.1371/journal.pntd.0008237 (2020).

:

REVIEWERS' COMMENTS:

Reviewer #1 (Remarks to the Author):

Change made are acceptable for publication to be warranted.